# Impact of Maternal Nutritional Supplementation during Pregnancy and Lactation on the Infant Gut or Breastmilk Microbiota: A Systematic Review

**DOI:** 10.3390/nu13041137

**Published:** 2021-03-30

**Authors:** Aneesa Z. Zaidi, Sophie E. Moore, Sandra G. Okala

**Affiliations:** 1Medical School, St George’s University of London, London SW17 0RE, UK; m1501166@sgul.ac.uk; 2Department of Women and Children’s Health, King’s College London, London SE1 7EH, UK; sandra.okala@gmail.com

**Keywords:** infant gut microbiota, breastmilk microbiota, microbiome, maternal nutritional supplementation, diet, pregnancy

## Abstract

Recent evidence indicates that maternal dietary intake, including dietary supplements, during pregnancy and lactation may alter the infant gut or breastmilk microbiota, with implications for health outcomes in both the mother and infant. To review the effects of maternal nutritional supplementation during pregnancy and lactation on the infant gut or breastmilk microbiota a systematic literature search was conducted. A total of 967 studies published until February 2020 were found, 31 were eligible and 29 randomized control trials were included in the qualitative synthesis. There were 23 studies that investigated the effects of probiotic supplementation, with the remaining studies investigating vitamin D, prebiotics or lipid-based nutrient supplements (LNS). The effects of maternal nutritional supplementation on the infant gut microbiota or breastmilk microbiota were examined in 21 and 12 studies, respectively. Maternal probiotic supplementation during pregnancy and lactation generally resulted in the probiotic colonization of the infant gut microbiota, and although most studies also reported alterations in the infant gut bacterial loads, there was limited evidence of effects on bacterial diversity. The data available show that maternal probiotic supplementation during pregnancy or lactation results in probiotic colonization of the breastmilk microbiota. There were no observed effects between probiotic supplementation and breastmilk bacterial counts of healthy women, however, administration of *Lactobacillus* probiotic to nursing women affected by mastitis was associated with significant reductions in breastmilk *Staphylococcal* loads. Maternal LNS supplementation during pregnancy and lactation increased bacterial diversity in the infant gut, whilst vitamin D and prebiotic supplementation did not alter either infant gut bacterial diversity or counts. Heterogeneity in study design precludes any firm conclusions on the effects of maternal nutritional supplementation during pregnancy and lactation on the infant gut or breastmilk microbiota, warranting further research.

## 1. Introduction

The human microbiota is a rich and diverse community of 10 to 100 trillion microbes including bacteria, archaea, parasites, fungi and viruses. Accumulating evidence indicates that the human microbiota, in particular the gut microbiota, which represents the largest reservoir of microorganisms in humans, plays an essential role in health and disease [1]. Changes in maternal diet during pregnancy and lactation have been linked to alterations in the maternal gut and breastmilk microbiota but also in the infant gut microbiota, with greater effects observed among breastfed infants compared to formula-fed infants [2]. These findings indicate that maternal nutrition is a key factor that shapes the infant gut microbiota, supporting the potential for maternal nutritional interventions to improve health outcomes and reduce disease risk. Recent advances in biomolecular technologies and bioinformatics have enabled a better understanding of the development of the gut microbiota and the complex commensal and symbiotic human-to-microorganism relationships [1]. These developments have highlighted three stages where the vertical transfer of microbial species from mother to infant and the establishment of the infant gut microbiota may occur; in utero, at birth and through breastfeeding, each route is briefly summarized below.

It has been traditionally believed that the uterus is sterile, with infant gut colonization occurring from birth, supported by recent evidence indicating that the placenta does not present a microbiota, though it may harbor microbial pathogens [3]. However, in contrast, some studies have detected microbial species in the placenta, amniotic fluid, fetal membranes and umbilical cord, providing evidence that vertical microbial transfer may be initiated in utero [4]. With the evidence remaining equivocal on the existence of a placental and/or fetal microbiota, the role of maternal diet during pregnancy on the development of the infant gut microbiota in utero remains uncertain. 

In contrast, there is clear evidence to indicate that infant gut colonization may occur at birth, with a study demonstrating that the mode of delivery influences the acquisition and composition of the infant gut microbiota, with differential outcomes between infants born vaginally compared to those born by caesarean section [5]. While vaginally born infants present a predominance of bacterial communities similar to those detected in the mother’s vagina including *Lactobacillus* and *Prevotella*, infants born by caesarean section, present a predominance of bacterial communities detected on the skin surface such as *Staphylococcus* and *Corynebacterium* [6]. This indicates that childbirth may be a critical time point for vertical transfer of microbial species and initiation of the human microbiota.

The process of labor, whereby the newborn is exposed to both vaginal and fecal bacterial species was widely accepted as the period for infant microbial colonization [7]. This theory can be challenged with studies demonstrating the presence of bacterial species including *Escherichia coli*, *Escherichia fecalis* and *Staphylococcus epidermis* in the infant meconium [8]. More recent studies have aimed to identify the origins of the meconium microbiota, and have established that it is more closely related to the amniotic fluid compared to the maternal fecal or vaginal microbiota [9]. This evidence again supports the hypothesis that infant gut colonization occurs in utero, indicating that maternal diet during pregnancy may indeed alter the infant gut microbiota. 

Finally, breastmilk and the process of breastfeeding have also been identified as key factors shaping the infant gut microbiota through both the breastmilk microbiota itself and breastmilk molecules such as human milk oligosaccharides (HMOs), anti-microbial factors and antibodies [10]. The breastmilk microbiota is a dynamic, complex unit, which evolves overtime and in close relation to the developing infant gut microbiota [10]. HMOs are abundant in the breastmilk, and are thought to selectively feed beneficial *Bifidobacterium* species in the gut of breastfed infants [11], resulting in a greater abundance of these bacterial species in breastfed infants compared to formula-fed infants [12]. Reductions in gut microbiota diversity or species richness have been reported among formula-fed infants and have been associated with increased future risks of atopic and metabolic diseases including, diabetes, obesity, and irritable bowel syndrome [13]. There is uncertainty whether maternal diet or maternal nutritional supplementation can alter the infant gut microbiota, and whether infant breastfeeding modulates these effects. To address this question, we conducted a systematic literature search in PubMed, Embase and Web of Science to summarize and evaluate the findings of these studies and determine whether reported effects differ by the type of nutritional supplement and infant feeding practice, and whether changes in the infant gut or breastmilk microbiota were associated with improved health outcomes.

## 2. Materials and Methods

This study was performed based on the PRISMA (Preferred Reporting Items for Systematic Reviews and Meta-Analyses) checklist [14] (Appendix A). The protocol has been registered in PROSPERO, registration number: CRD42020167909.

### 2.1. Search Strategy

A systematic literature search was conducted using PubMed, Embase and Web of Science using combinations, variations and synonyms of the terms “maternal”, “supplement*”, “breastmilk”, “microbiota”, “infant” and “gut”, to identify relevant studies published up to February 2020. The full search strategy is given in the Appendix A.

### 2.2. Inclusion and Exclusion Criteria

Randomized controlled trials (RCTs), quasi-randomized controlled studies and cohort studies were included in the systematic review. Conference abstracts and unpublished reports were only included if they provided sufficient information regarding the study design, participant characteristics and stated results, however, single case reports, systematic reviews and expert opinions were excluded. Only studies published in the English language were included. Studies of poor quality of evidence were excluded from the narrative synthesis after full-text screening if there was concern about the accuracy and reproducibility of the data provided. The participants included in the studies were women aged 18 years or older who were randomized to receive nutritional supplements during pregnancy and/or lactation. All participants included must have provided breastmilk samples and/or stool samples from their infants. Participants were not excluded based on their health status, allowing for both infants and women with health conditions such as mastitis, allergy or eczema, clinically diagnosed during the study or with a family history to be included. Nutritional supplements were defined as concentrated sources of nutrients or other substances that have a nutritional or physiological effect and enhance the normal diet [15]. Thus, the review included interventions with any form of nutritional supplementation taken during pregnancy; from conception to delivery and/or postnatally; during lactation. Any studies examining the effects of maternal diet or dietary food intake were excluded. Nutritional supplements may have been self-administered or prescribed and given at any dose, duration and frequency. Included studies had a comparator group of women who either did not consume a supplement during pregnancy or lactation, or who received a placebo defined as a compound not containing the active nutritional supplement ingredient and given in the same condition as the supplement. 

The studies eligible for inclusion in the review provided information on the effects of maternal nutritional supplementation during pregnancy or lactation on the infant gut microbiota and/or the breastmilk microbiota. The outcomes measured comprised any alterations of the composition, diversity, function and abundance or lack of alterations in the infant gut microbiota and/or breastmilk microbiota. Details on additional outcomes were also collated and included any reported differences in clinically diagnosed health conditions and infant feeding practices during the study. 

### 2.3. Data Extraction and Analysis

Two of the authors (AZ and SO) reviewed all articles and performed the study selection independently according to the inclusion and exclusion criteria. Disagreements were resolved by discussion to reach a consensus. The data were extracted and stored on a data extraction table. Extracted information included: title, year, study design, geographical location, study population; including the sample size and participant demographics, aims and/or objectives, intervention and control; including the dose, duration and frequency of the supplement, study methodology and study results.

Studies were initially appraised individually, before comparing and summarizing the findings. This allowed for a descriptive synthesis of data, followed by a review by themes. Generated tables included a summary of the participant characteristics and the main results for all studies included in the review. The results of all studies were compared to assess the impact of maternal nutritional supplementation on the infant gut or breastmilk microbiota. The studies were then examined for links between maternal nutritional supplementation and health outcomes and whether reported effects were influenced by infant feeding practice. 

### 2.4. Risk of Bias and Quality of Information Assessment

The Cochrane Collaboration’s tool [16] was used to assess the risk of bias in the selected studies. Each type of bias was graded as low, high or unclear based on provided evidence. The GRADE approach [17] was also used to determine the overall quality of each study. This grades each study as either high, moderate, low or very low quality of evidence based on the degree of confidence in the estimate of effect. The approach incorporates numerous factors including the study design, risk of bias, indirectness of evidence and inconsistency or imprecision of results to determine the quality of each study.

### 2.5. Statistical Analysis 

Studies included in the review differed widely by type of supplements, duration of treatment and population characteristics, therefore a meta-analysis could not be performed on the primary outcomes. However, one sub-group meta-analysis could be performed on whether maternal probiotic supplementation during pregnancy and lactation results in the colonization of the breastmilk microbiota by the supplemented probiotic due to homogeneity across these studies. This meta-analysis was performed using RevMan 5. An odds ratio (OR) and 95% confidence intervals (CI) were calculated for individual studies and then a combined estimate OR for the pooled results was generated using a random-effects method; DerSimonian and Laird inverse variance.

## 3. Results

### 3.1. Study Selection and Characteristics

The search from selected databases yielded 965 publications and two studies were identified through backward chaining. After the removal of 212 duplicates, 753 articles were screened based on their title and abstract, 722 studies were excluded, resulting in 31 studies assessed for eligibility. After full-text screening, two studies were eliminated due to inadequate quality of evidence, resulting in 29 studies included in the systematic review (Figure 1).

Selected studies were all randomized controlled trials published between 2005 and 2020, including 22 (76%) studies published in the last 10 years. The studies largely took place in high-income countries, with the exception of two studies which were both conducted in Malawi; a setting with a high prevalence of maternal and infant malnutrition [18,19]. The study locations covered all six WHO regions and 19 (66%), were conducted in Europe (Figure 2). Ethical approval was sought in 27 of the studies, and four studies [20,21,22,23] failed to mention ethics, leaving their ethical approval status unclear. 

### 3.2. Population Characteristics

All studies included women of reproductive age (18 to 45 years old), however, the health status and participant demographics differed across the studies. A total of 18 out of the 29 studies involved women or infants with high risk of atopic disease [23,24,25,26,27,28,29,30,31,32,33,34,35,36,37,38,39,40]. Of these, six studies were part of allergy prevention trials [23,25,26,28,29,33], one study specifically did not exclude women with a history of allergic disease [31] and the remainder, recruited participants who either had a diagnosis of the atopic condition before recruitment or had a positive family history for the condition (Figure 3). Conditions included were asthma, eczema, hay fever, food allergy and allergic rhinitis. Three studies involved lactating women with health conditions related to breastfeeding [41,42,43]. Two studies included women suffering from acute breastfeeding illness [42,43]; women with clinical symptoms of *Staphylococcal* mastitis, [42], and women with painful breastfeeding not related to lactational mastitis [43], respectively (Figure 3). One paper included healthy women with a history of lactational mastitis after at least one previous pregnancy [41]. In eight studies, selected participants were healthy pregnant women with no stated health conditions or pregnancy complications [18,19,20,21,22,44,45,46] (Figure 4). 

Regarding the type of maternal nutritional supplementation, 23 studies administered probiotic supplementation to the intervention arm [20,21,22,24,25,26,27,28,30,32,33,34,35,36,37,38,39,40,41,42,43,44,45]. The intervention regime and strain of the probiotic varied greatly, with 10 studies administering a multi-strain preparation [20,25,26,27,28,32,37,38,42,45]. There were 16 different bacterial species present in the probiotics, however, the most abundant species present was a single strain preparation of *Lactobacillus rhamnosus*. There were two studies that examined the effects of prebiotic supplementation [31,46], while a further two studies supplemented women with lipid-based nutrient supplements (LNS) [18,19]. The remaining two studies investigated the effects of maternal nutritional supplementation with vitamin D [23,29].

All 29 studies were assessed using the GRADE approach and ranked as high, medium, low or very low study quality (Table 1). This approach incorporates the risk of bias assessment (Appendix A), to make an overall judgement on the study quality. 

The reported impact of maternal nutritional supplementation on the infant gut (Table 2) and breastmilk microbiota (Table 3) are described below. The studies investigating the effects of probiotic supplementation will be discussed first, followed by those on the effects of prebiotic, LNS and vitamin D supplementation. 

### 3.3. Probiotics

#### 3.3.1. Infant Gut Microbiota

There were five studies that measured whether maternal probiotic supplementation administered during pregnancy and lactation resulted in the probiotic colonization of the infant gut, with all studies demonstrating significant results for the outcome assessed [26,27,34,37,40]. However, a meta-analysis could not be performed due to heterogeneity in the probiotic composition, timing of regimen and the presentation of the results. In a large Norwegian RCT study (*n* = 415), women were administered a multiple strain probiotic consisting of *Lactobacillus rhamonosus*, *Lactobacillus acidolphilus* and *Bifidobacterium animalis* subspecies *lactis* or a placebo from 36 weeks of gestation up to 3 months postnatally while breastfeeding. The results of this trial showed that only the *Lactobacillus rhamonsus* bacteria colonized the infants gut of women who were adminstered the probiotic as observed with an increased abundance of this bacteria in their infants stool samples at 10 days and 3 months of age compared to the control group (both, *P* < 0.005). However, at 1 or 2 years of age, there were no longer any significant differences in the abundance of the administered probiotic bacteria in their infants. [26]. Similarly, in a 3-arm RCT conducted in New Zealand (n=600) where women (from 35 weeks gestation until 6 months post-partum if breastfeeding) and their infants (from birth to 2 years) were administered *Lactobacillus* rhamonsus, *Bifidobcaterium animalis* subspecies *lactis* or a placebo, only *Lactobacillus rhamnosus* colonized the infants gut in the probiotic group with detection levels ranging from 71.5% of the samples at 3 months to 62.3% at 24 months of age [34]. In another 3-arm trial conducted in New Zealand (*n* = 474) with the same treatment groups and administration regimen as the aforementioned study, *Lactobacillus rhamonsus* was significantly more abundant at all measurement time points from birth to 24 months of age in the probiotic group compared to the control group [40]. A study conducted in the Netherlands found that administration of a multiple strain probiotic to pregnant women during the last 6 weeks of pregnancy and to their infants up to one year of age, containing *Bifidobacterium* and *Lactococcus* species resulted in a significant increase in *Bifidobacteria* counts at one month of the infant’s age (*P* = 0.0033) and *Lactococcus lactis* counts at 2 weeks of the infant’s age (*P* = 0.001) [37]. This can be contrasted with a separate study, conducted in Finland, whereby women were supplemented with a multiple strain probiotic two months before delivery and two months after delivery or until they stopped breastfeeding. This study showed no evidence of infant gut colonization during supplementation (*P* = 0.11 for *Bifidobacterium* and *P* = 0.40 for *Bifidobacterium longum*), however, after cessation of the supplement, evidence of probiotic colonization in the infant gut couple be detected (*P* = 0.043 for *Bifidobacterium* and *P* = 0.023 for *Bifidobacterium longum*) [27].

The impact of maternal probiotic supplementation on bacterial counts present in the infant’s gut varied across studies. Three out of five studies found that probiotic supplementation administered during pregnancy and/or lactation increased *Bifidobacteria* counts in the infant’s gut. In a study conducted in Sweden (*n* = 232), women were randomized to receive 1 × 10^8^ CFU of *Lactobacillus* probiotic or a placebo during the last 4 weeks of pregnancy, thereafter, the infants continued with the same study product from birth to one year of age. In this study, significantly higher counts of *Bifidobacteria* were detected in infants in the probiotic group compared to the control group, with highest counts measured at 5 to 6 days of age (82% vs. 20%, *P* < 0.001) [24]. Likewise, in an Australian study (*n* = 122) using the same concentration of a *Lactobacillus* probiotic, *Bifidobacterium longum* species were detected more frequently in the intervention group than in the control group (82% vs. 61%; *P* < 0.01; prevalence ratio, 1.35; 95% CI 1.06–1.72) [33]. In a study conducted in Finland where women were randomized to receive either supplementation with multiple strain probiotics or a placebo 2 months prior and 2 months after delivery, no significant differences in *Bifidobacteria* counts were detected in infants stool samples between the groups at one month of age. However, at 6 months of age, a significant difference was observed (*P* = 0.043 for *Bifidobacterium* genus and *P* = 0.023 for *Bifidobacterium longum*) [27]. Conversely, in two other studies conducted in Finland where women were administered 1 × 10^10^ CFU of *Lactobacillus rhamnosus* probiotic during pregnancy and lactation, *Bifidobacteria* counts detected in infants stool samples at 6 months were comparable between the supplement and the control groups (*P* = 0.70, [36] and *P* = 0.145, [35]).

There were two studies that measured whether probiotic supplementation administered during pregnancy and lactation altered *Lactobacilli* counts in the infant’s gut. In a study conducted in Italy (*n* = 35), where women were administered a multiple strain probiotic or a placebo during late pregnancy and lactation, higher *Lactobacilli* counts were detected in neonatal faecal samples from the probiotic group compared to the control group (*P* < 0.05) [20]. Conversely, in a study conducted in Finland (*n* = 132), where women were administered *Lactobacillus rhamnosus* or a placebo before delivery to 6 months post-delivery, no differences in *Lactobacilli/Enterococci* counts were observed at 6 months of age between the probiotic and control groups, but at 2 years of age lower *Lactobacilli/Enterococci* counts were observed in the probiotic group compared to the control group (*P* = 0.011) [35]. 

There were four studies examining the effects of maternal probiotic supplementation administered during pregnancy and/or lactation on the diversity of the infant gut microbiota [26,30,34,37]. In a study conducted in New Zealand (*n* = 600), with women receiving either a single strain probiotic or a placebo during pregnancy and lactation, no significant differences in infant gut microbiota were observed as measured by Bray–Curtis distance (*P* > 0.05) for the beta diversity, or Shannon’s index for the alpha diversity [34]. Similarly, in an Australian study (*n* = 98), with women receiving either *Lactobacillus rhamnosus* or a placebo during pregnancy, no significant difference in the mean number of peaks was detected in infant faecal samples at one week of age between the groups as measured by terminal restriction fragment length polymorphism using restriction enzymes [30]. In a study conducted in the Netherlands (*n* = 123), with women in the last 6 weeks of pregnancy and their infants in the first year of life receiving either a multiple strain probiotic or a placebo, significantly higher *Bacteroidetes* and *Proteobacteria* diversity were observed in the placebo group at 2 weeks and 2 years of age compared to the probiotic group (*P* < 0.05) [37].

#### 3.3.2. Breastmilk Microbiota

There were seven studies that investigated whether probiotic supplementation during pregnancy and lactation resulted in probiotic colonization of the breastmilk [21,24,33,38,39,41,45]. A total of six studies observed the outcome assessed, with only one study reporting no detection of the probiotic in the breastmilk of supplemented women [39]. A meta-analysis was also performed due to homogeneity across these studies regarding the intervention and outcome measured. The results demonstrate that maternal probiotic supplementation may result in colonization of the breastmilk microbiota (OR 5.93; 95% CI 2.32-15.20, *P* = 0.0002) (Figure 5).

There were three studies that included lactating women either at risk of, or suffering from lactational mastitis [41,42,44]. Overall, all three studies demonstrated that maternal supplementation with *Lactobacillus* probiotic administered during pregnancy and/or lactation reduced *Staphylococcal* counts in the breastmilk of supplemented women. In a Spanish RCT (*n* = 108), where women were administered 9log_10_ CFU of *Lactobacillus salivarius*, there was a small but significant difference in mean *Staphylococcal* bacterial counts in the milk samples from healthy women who had received the probiotic compared to healthy women in the placebo group (0.19 [95% CI 0.09–0.30] log_10_ CFU/mL) [41]. The remaining two studies found significantly lower levels of *Staphylococcus* subspecies in the breastmilk of women treated with the probiotic during lactation [42,44]. Of these, a further study conducted in Spain (*n* = 291), administered 3 × 10^9^ CFU of *Lactobacillus fermentum* for 16 weeks to the intervention group. This group observed a reduction by 48%; (*P* = 0.013) of *Staphylococcus* subspecies amongst healthy women and a greater reduction by 58% (*P* = 0.065) in women with diagnosed mastitis. Overall, at the end of the intervention, the bacterial load was lower in the breastmilk of women in the probiotic group compared to the control group (*P* = 0.025) [44]. A similar result was reported in a smaller Spanish study (n = 20) where the intervention group was administered 200mg of 10log_10_ CFU of a multiple strain probiotic for 4 weeks. At the end of the study, the intervention group had a bacterial count of 2.96 log_10_ CFU/mL which was significantly lower than in the control group at 4.79 log_10_ CFU/mL (*P* = 0.002) [42]. Another Spanish study (*n* = 148), included women suffering from breast pain and elevated milk bacterial counts, not associated with acute mastitis and also reported a significant decrease in *Staphylococcus* load in the intervention compared to the control group (*P* = 0.045) [43]. Notably, treatment with 3 × 10^9^ CFU of *Lactobacillus fermentum* for 3 weeks significantly reduced *Staphylococcus* load in the breastmilk (*P* = 0.011). 

A Norwegian RCT (*n* = 252), found no significant effects of supplementation with a multiple strain probiotic administered 4 weeks before delivery to 3 months after birth on the alpha or beta diversity of the breastmilk microbiota [38]. However, this study indicated that the supplement may have a positive influence on the stability of the breastfeeding-associated microbiota (RR of stable breastfeeding-associated microbiota after probiotic supplementation: 2.37, 95% CI 0.94–5.97, Fisher’s exact *P* = 0.050).

### 3.4. Prebiotics

The effects of prebiotic supplementation on infant bacterial counts were investigated in two studies conducted in Japan and Germany [31,46]. In the study based in Germany, with women receiving prebiotics; galactooligosaccharides (GOS) and long-chain fructooligosaccharides (lcFOS) or a placebo from 25 weeks’ gestation until delivery (*n* = 48), no significant differences in percentages of *Bifidobacteria* present in the infant stool at 5, 20 and 182 days of age between the groups were observed [46]. Conversely, in the study based in Japan (*n* = 84), with women supplemented during pregnancy and lactation with FOS or a placebo, a borderline significant difference in the number of *Bifidobacteria* in the intervention group compared to the control group (*P* = 0.50), and an increased number of *Bifidobacterium longum* species were detected (*P* = 0.01) [31]. 

### 3.5. Lipid-Based Nutrient Supplements

The effects of LNS administered during pregnancy and lactation on the infant gut microbiota were assessed in two large RCTs conducted in Malawi [18,19]. In the first study, 869 women were supplemented with either multiple micronutrients (MMN), iron and folic acid (IFA) tablet-based supplements or LNS during pregnancy until 6 months postpartum. The infants in the LNS group received LNS from 6 to 18 months of age, whilst infants in the control group did not receive any supplements during the study. A higher alpha diversity (Shannon index *P* = 0.032), Pielou’s evenness function (*P* = 0.043), alongside a trend for increased species richness (*P* = 0.08) were observed in the LNS group in infants at 18 months of age compared to the IFA and MMN groups but no differences in beta diversity were detected at any age [18]. Conversely, in the second study (*n* = 631), women were randomly assigned to the intervention group, supplemented with LNS or MMN during pregnancy and until 6 months postpartum, or the control group, supplemented with only IFA tablet-based supplements. The infants born were thereafter supplemented with either LNS or no intervention between 6 and 18 months of age depending on which arm of the trial their mother was assigned to. Overall, this study demonstrated that there were no significant effects of LNS on the infant gut microbiota diversity and maturity in samples collected from 6 to 30 months of age [19]. 

### 3.6. Vitamin D

The impact of vitamin D supplementation during pregnancy on the infant gut microbiota were assessed in two RCTs conducted in Denmark (*n* = 880) and the USA (*n* = 736) [23,29]. In both studies, there were no significant differences on either the alpha diversity or beta diversity in infants gut microbiota within the first year of life between the groups receiving vitamin D supplementation or a placebo. 

### 3.7. Additional Outcomes

#### 3.7.1. Health Outcomes

There were four studies that investigated whether maternal probiotic supplementation administered during pregnancy and lactation reduced the development of eczema in high-risk infants [34,37,39,40]. A study conducted New Zealand (*n* = 474) showed that whilst maternal supplementation with *Bifidobacterium animalis* subspecies during pregnancy and lactation was not associated with a reduced risk of eczema in infants, supplementation with *Lactobacillus rhamnosus* was associated with a significant reduced risk of eczema (hazard ratio [HR], 0.51; 95% CI 0.30–0.85) compared to those in the control group (*P* = 0.01) [40]. A beneficial effect of maternal supplementation with a probiotic mixture on the development of eczema was also reported in a study conducted in the Netherlands, with a lower prevalence of eczema in the first 3 months of life in the probiotic group compared to the control group (12% vs. 29%) [37]. In contrast, two other studies conducted in Australia (*n* = 600) and New Zealand (*n* = 423) did not observe effects of maternal supplementation with probiotics during pregnancy and lactation on the prevalence of eczema [34,39]. 

There were four studies [41,42,43,44] that reported whether probiotic supplementation during lactation either improved or prevented symptoms of mastitis. All studies observed a reduction in either symptoms or bacterial counts amongst women in the intervention groups. In a Spanish RCT including 108 women with a history of mastitis, there were significantly lower level of bacterial counts in women with subacute and acute mastitis who were administered 9log_10_ CFU of *Lactobacillus salivarius* probiotic from 30 weeks of pregnancy until delivery, compared to those who were administered the placebo [41]. Similarly, a separate Spanish RCT (*n* = 291), observed significantly lower incidence rates of mastitis in women administered 3 × 10^9^ CFU of preventative *Lactobacillus fermentum* probiotic for a 16 week period compared to those in the control group (IR = 0.130 vs. IR = 0.263, *P* = 0.021) [44]. The odds of experiencing breast pain amongst women in the probiotic group were also significantly lower than women treated with the placebo, (OR 0.65; 95% CI 0.44–0.97) [44]. In a Spanish study (*n* = 148), women with painful breastfeeding and elevated breastmilk bacterial counts showed that administration of three different concentrations of a *Lactobacillus fermentum* probiotic for 3 weeks all improved breast pain scores and decreased bacterial counts [43]. 

Multiple studies measured cytokines and antibodies in the breastmilk as a proxy for health status, with two studies out of four observing probiotic supplementation altering levels. In a small Italian study of 35 healthy pregnant women probiotic supplementation during pregnancy and lactation was associated with increased levels of TGF- β1 in the colostrum and IL-10 levels in the mature milk [20]. In a Finish RCT (*n* = 96), infants at high risk of allergic disease born to supplemented mothers who had been exclusively breastfeeding for at least 3 months had higher total numbers of IgM, IgA, and IgG compared to those in the placebo group [36]. Conversely, in a large study conducted in New Zealand study of 474 women with a history of allergic disease, probiotic supplementation during pregnancy and lactation did not affect the levels of TGF-β1, TGF-β2 or IgA in their breastmilk [40]. 

#### 3.7.2. Impact of Feeding Practice

There were three studies that subcategorized mother-infant pars by mode of feeding. In a large Finish RCT (*n* = 1223), women were administered a multiple strain probiotic or a placebo during pregnancy and lactation. In this study breastfeeding was associated with an increased abundance of *Lactobacilli* and *Bifidobacteria* by 100% and 29%, respectively, in infant’s faecal samples from the probiotic groups whereas a slight, but significant decrease in the total abundance of *Bifidobacteria* was reported in formula fed infants (7%, *P* < 0.0001) from the same group [32]. Likewise, in another study conducted in Finland (*n* = 96), greater effects of probiotic supplementation during pregnancy and lactation were observed in infants who were exclusively breastfed for at least 3 months compared to formula fed infants in the probiotic group with significantly higher faecal *Bifidobacterium* and *Lactobacillus/Enterococcus* counts at 6 months reported in EBF infants (*P* < 0.0004 and *P* = 0.10) [36]. Conversely, in a RCT conducted in Sweden (*n* = 232), exclusive breastfeeding was associated with lower levels of the probiotic *Lactobacillus reuteri* in the infant gut microbiota compared to formula-fed infants (*P* < 0.001) [24]. The study also reported lower counts of *Clostridium difficile* at 3 and 6 months (*P* < 0.001) in exclusively breastfed versus formula-fed infants.

## 4. Discussion

The present systematic review suggests that maternal nutritional supplementation during pregnancy and lactation can impact on the infant gut microbiota or breastmilk microbiota, but with varying effects according to participant characteristics, type of supplement administered, and outcome measured. Despite the heterogeneity of the findings, this review highlights the potential to improve the health of young infants through maternal nutritional supplement-based interventions during pregnancy and lactation, although further research is warranted. 

Our review yielded 29 randomized controlled trials of relevance, with the majority (23/29) focused on probiotic supplementation. Of these, most studies reported that maternal probiotic supplementation during pregnancy and lactation resulted in the probiotic colonization of the infant gut [26,27,34,37,40] and breastmilk microbiota [21,24,33,38,39,41,45]. These findings were confirmed by the results of our meta-analysis including seven studies which showed that maternal probiotic supplementation during pregnancy and lactation result in the probiotic colonization of the breastmilk with the bacterial species administered. Although a meta-analysis could not be performed across all the trials identified due to high heterogeneity between studies, in a descriptive synthesis, we observed comparable results indicating that probiotic supplementation during pregnancy and lactation and in infancy were associated with the probiotic colonization of the infant gut. These findings are in line with the entero-mammary hypothesis which suggests that dendritic cells may capture bacteria cells from the maternal gut lumen, transport them to the mammary gland, thereby enabling their transfer through breastfeeding [47]. In the five studies that reported infant gut colonization by the probiotic, women were administered the probiotic across pregnancy and lactation, therefore it is unclear as to whether the probiotic colonization of the infant gut begun in utero or from birth. The principles of evolutionary biology suggest that the intrauterine environment includes protective maternal barriers that may limit microbial transfer during fetal life [48]. These barriers include placental trophoblasts that recognise pathogens via Toll-like receptors (TLRs) and Retinoic acid-inducible gene I (RIG-I)-like receptors (RLRs), thereby activating antimicrobial signaling pathways ensuring a stable environment for the vulnerable fetus [49]. In four out of five studies, the probiotic prevalence was the highest in the infant’s gut within the first few months of life (26,34,37,40), before steadily declining, perhaps due to competition from other evolving bacterial communities [50]. Some studies suggest that bacterial colonization may occur immediately after birth and that the gut undergoes continuous development to establish a mature microbiota by 3 years of age [51].

Although maternal probiotic supplementation during pregnancy and lactation appeared to alter infant gut bacterial counts, no definitive conclusions could be drawn due to broad differences in the methodology used to measure the outcome of interest with some, particularly measuring counts of bacterial genus and others the counts of specific species within the genus. However, there is some evidence suggesting that probiotic supplementation during pregnancy and lactation may increase *Bifidobacteria* counts in the infant’s gut, with three out of five studies reporting this finding [24,27,33]. The most abundant bacterial species present in the human milk of healthy women is thought to be *Staphylococcal* bacteria [52], followed by *Lactobacilli* and *Bifidobacteria* [53], therefore, the natural high abundances of bacteria present in the human milk may limit the ability to alter bacterial proportions.

The effects of maternal probiotic supplementation on the bacterial counts present in the breastmilk microbiota was dependent on the health status of participants. Maternal supplementation with *Lactobacillus* probiotic during pregnancy and/or lactation reduced *Staphylococcal* counts in the breastmilk of supplemented women at risk of or suffering from mastitis [41,42,44] or with breast pain not associated with mastitis [43]. Although these studies provided high-grade quality of evidence, some limitations should be noted; with one study presenting a small sample size (*n* = 20) [42], and all studies using varying *Lactobacillus* probiotic subspecies of different concentrations, limiting the interpretation of the findings across studies.

Maternal supplementation during pregnancy and lactation with prebiotics including GOS and lcFOS was found to have conflicting yet limited effects on infant gut *Bifidobacteria* counts, with one study reporting no significant differences while another study showed a borderline increase in the intervention group compared to the control group [31,46]. Prebiotics have the potential of modifying the gut microbiota, however these modifications occur at the level of individual prebiotic strains and are therefore not easily predicted [54]. There is also evidence to suggest that responses to prebiotics are highly influenced by gut pH [55], which also may provide an explanation for the uncertainty surrounding the impact of maternal prebiotic supplementation on the infant gut microbiota in this systematic review.

This review found that nutritional supplementation with probiotic [26,30,34,37,38] or vitamin D [23,29] had very limited effects on the alpha and beta diversity of the breastmilk and infant gut microbiota. However, a study conducted in Malawi indicated that administration of LNS during pregnancy and lactation increased infant gut microbiota diversity at 6 months of age [18], although a further study in the same setting did not observe this effect [19]. LNS contain essential fatty acids as well as other micronutrients including vitamin D and calcium [56]. The gut microbiota is involved in the metabolism of lipids and is thus predicted to be affected by lipid supplementation [57]. Furthermore, there is evidence to suggest that high-fat diets rich in long-chain fatty acids modulate the gut microbiota resulting in dysbiosis, inflammation and even an increased risk of obesity and metabolic syndrome [58]. This evidence suggests that there is rational for using LNS to improve the infant gut microbiota and even minimise disease risk. However, currently, there is limited evidence to support this hypothesis, leaving the impact of maternal LNS supplementation on the infant gut microbiota unclear.

Maternal probiotic supplementation was found to generally improve clinical symptoms of mastitis and reduce atopic disease in infants. Systematic reviews that purely investigate whether maternal supplementation reduced atopic disease in the infant concluded that probiotics administered to pregnant or breastfeeding women were associated with a reduction in risk of eczema in infants [59,60]. However, the authors also highlighted that the imprecision of the estimated pool effects and likelihood of bias were high [59], limiting the interpretation of these findings. Likewise, clear conclusions could not be drawn from our systematic review as there were only four studies investigating this outcome and only two of these reported an association between maternal supplementation during pregnancy and lactation and reduction in eczema in infants. Outcomes were also measured using different tools and often relied on clinical judgement, rather than quantitative measurements, further limiting the quality of evidence. Future studies would, therefore, benefit from standardized objective measurement tools.

In the three studies that subcategorized mother-infant pairs by mode of feeding, two reported greater differences in bacterial counts in the gut microbiota of exclusively breastfed infants compared to formula-fed infants both born to supplemented mothers. This result is in line with a previous systematic review that demonstrated the microbial composition differs between non-exclusively breastfed versus exclusively breastfed infants [61]. In the current review, only three of the included studies categorized participants by infant feeding practice precluding any conclusions on whether feeding practice modulates the effects of maternal nutritional supplementation on either the infant gut or breastmilk microbiota.

Recent studies show that maternal nutritional supplementation during pregnancy and lactation may modulate the infant gut and/or breastmilk microbiota [62] and several reviews have been conducted to assess whether maternal probiotic supplementation can improve infant health outcomes, namely reduce the risk of atopic disease [59,63,64,65]. However, to the best of our knowledge there are no reviews investigating the effects of maternal nutritional supplementation on the infant gut or breastmilk microbiota. Overall, this systematic review suggests that maternal nutritional supplementation may modulate the infant gut or breastmilk microbiota. Probiotic supplementation during pregnancy and lactation was found to colonize both the infant gut and breastmilk microbiota. There is also some evidence to suggest that probiotic supplementation may modulate infant gut bacterial counts and reduce bacterial counts in the breastmilk of women with mastitis.

### Strengths and Limitations of the Review

This systematic review is strengthened by multiple aspects of the methodological approach. The systematic search strategy carried out by two independent reviewers ensured all relevant literature was identified and evaluated. This was accompanied by a broad inclusion criterion with no restriction on the health status of participants or the intervention regimen, allowing for all studies in this area of research to be included. The eligible studies were then assessed for their risk of bias and an overall judgement was made on their quality of evidence and the vast majority of included studies were of good quality of evidence. There were 20 studies in this systematic review that used the gold standard technique; real-time (RT) polymerase chain reaction (PCR) and primers targeting specific 16S variable gene regions [66], to evaluate the bacteria present in the microbiota, with the remaining nine studies utilizing a variety of alternative analytical methods (Appendix A). Future studies assessing the infant gut and/or breastmilk microbiota would therefore benefit from using this method of assessment to ensure a consistency in methodology across all studies. 

The main limitation of this review is the lack of homogeneity between studies concerning the health status of participants and the composition and duration of nutritional supplement administered, and the method used to measure the outcomes assessed. This is largely due to the limited studies available on this area of research. A broad inclusion criterion was decided following from the limited number of records identified on preliminary searches with stricter inclusion criteria regarding both the participants included and the nutritional supplements administered. There was concern that the varying health status of participants across our selected studies may bias our results, with studies demonstrating differences in the proportions of bacteria between the healthy population and patients with allergic disease including atopic dermatitis [67]. In addition, these conditions are associated with an imbalance of T helper (Th) 1/Th2 cells and higher proportions of cytokine secretion. These immunological pathways can interact with probiotics by either inducing immune activation signaling or trigger tolerance signaling through the production of cytokines. Overall this may cause alterations in the gut microbiota, limiting the comparison between studies of healthy pregnant women versus studies of women either suffering from or at increased risk of atopy in our systematic review. The majority of the studies in our systematic review; 18 out of 29 studies involved women or infants with high risk of atopic disease [23,24,25,26,27,28,29,30,31,32,33,34,35,36,37,38,39,40], and three studies specifically involved women with breast-related illness [41,42,43,44], meaning in eight studies selected participants were healthy pregnant women with no stated health conditions [18,19,20,21,22,44,45,46]. This means that the comparability between the 18 studies of participants with high risk of atopic disease and the eight studies of healthy participants mentioned may be limited. The heterogeneity amongst our selected studies meant that a meta-analysis could not be performed on all the main outcomes and may preclude any definitive conclusions. Nonetheless, this systematic review presents to the best of our knowledge the first collation of evidence on the effects of maternal nutritional supplementation during pregnancy and lactation on the infant gut or breastmilk microbiota.

## 5. Conclusions

This systematic review indicates that maternal probiotic supplementation during pregnancy and lactation may result in probiotic colonization of the infant gut and breastmilk microbiota. Furthermore, there is evidence that maternal probiotic supplementation during pregnancy and/or lactation may alter bacterial counts in the infant gut microbiota and reduce *Staphylococcal* counts in the breastmilk of women with mastitis. The review highlighted that knowledge and understanding of this area of research are promising but still limited to form definitive conclusions. Recommendations are therefore directed towards the need for further, high-quality RCTs examining the role of maternal nutritional supplementation during pregnancy and lactation on the infant gut and/or breastmilk microbiota.

## Figures and Tables

**Figure 1 nutrients-13-01137-f001:**
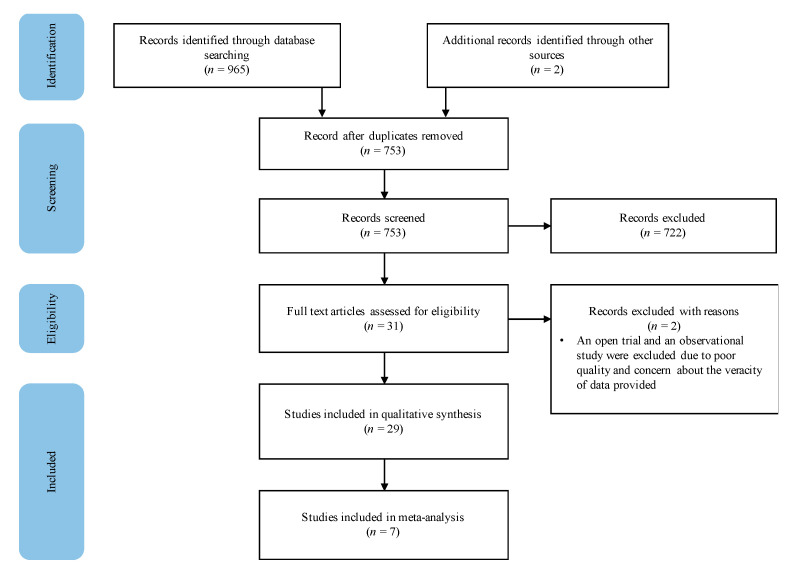
Study selection procedure.

**Figure 2 nutrients-13-01137-f002:**
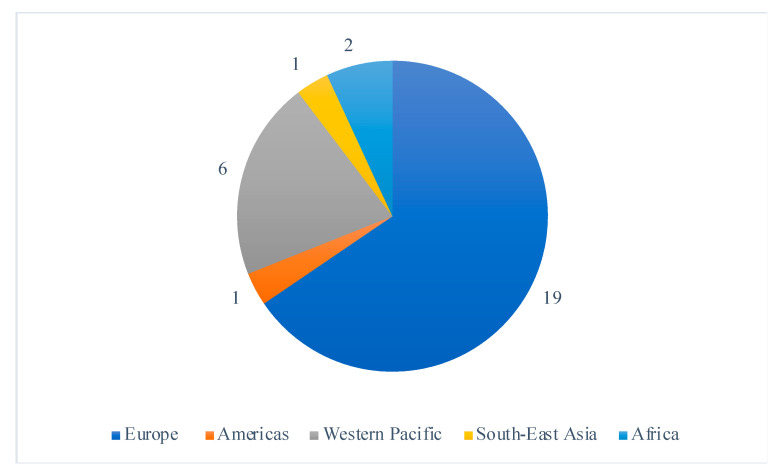
Distribution of the included study’s location sites by WHO region.

**Figure 3 nutrients-13-01137-f003:**
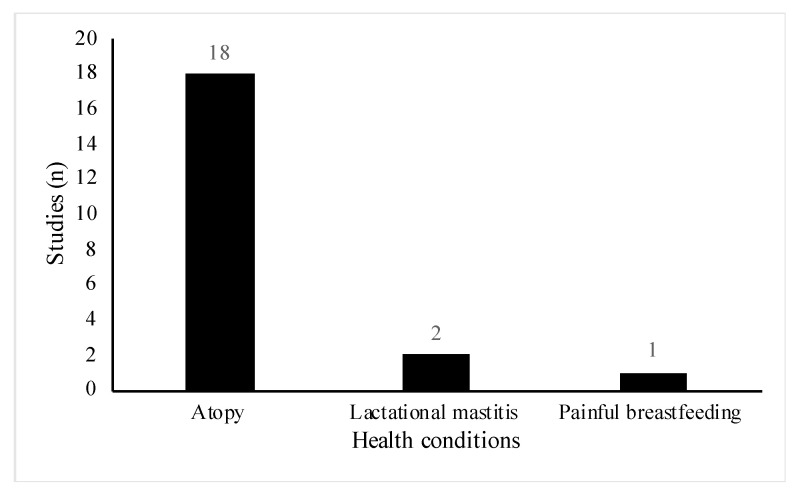
Distribution of specific health conditions among included studies (*n* = 21).

**Figure 4 nutrients-13-01137-f004:**
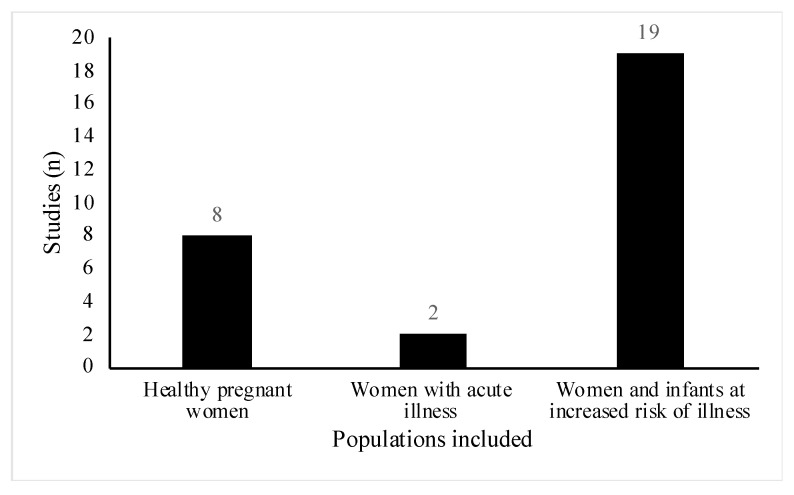
Distribution of healthy pregnant women compared with pregnant women and/or their infants at increased risk of or suffering from an illness in included studies (*n* = 29). Of note, healthy pregnant women include women where studies do not mention any specific health conditions.

**Figure 5 nutrients-13-01137-f005:**
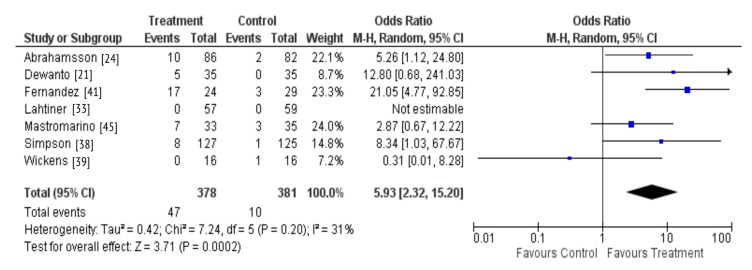
Probiotic supplementation and colonization of the breastmilk microbiota.

**Table 1 nutrients-13-01137-t001:** Participant characteristics in included studies (*n* = 29).

Reference	Country	Grade	Sample Size (*n*)	Participants	Health Conditions	Outcome Assessed (Breastmilk, Infant Gut Microbiota or Both)
Abrahamsson et al. [24]	Sweden	High	232	At least one family member with an allergic disease	Allergic disease	Both
Avershina et al. [25]	Norway	Medium	415	Part of the Probiotics in Prevention of Allergy Among Children in Trondheim study (ProPACT). Pregnant women ≤36 weeks’ gestation, planning exclusive breastfeeding (EBF) for 3 months	Not restricted to a family history (FH) of allergic disease *	Infant gut microbiota
Baldassarre et al. [20]	Italy	Medium	35	Healthy pregnant women	Nil	Both
Dewanto et al. [21]	Indonesia	Medium	70	Healthy pregnant women in their 3rd trimester, not receiving antibiotics and planning EBF for at least 3 months	Nil	Breastmilk microbiota
Dotterud et al. [26]	Norway	High	415	See Avershina et al. [25]		Infant gut microbiota
Fernández et al. [41]	Spain	High	108	Healthy pregnant women with a history of lactational mastitis	Lactational mastitis	Breastmilk microbiota
Fonollá Joya et al. [22]	Spain	Low	291	Breastfeeding women	Nil	Both
Grönlund et al. [27]	Finland	High	80	High-risk allergy family; mother had clinical symptoms of allergy with prick-test-proven reactivity against allergens. Planning EBF for a minimum of 4 months, followed by partial/EBF for a further 2 months	Allergic disease	Infant gut microbiota
Grześkowiak et al. [28]	Finland and Germany	Medium	79	Part of an ongoing allergy prevention study and planning to EBF for a minimum of 4 months, followed by partial/EBF for a further 2 months.	Nil	Infant gut microbiota
Hjelmsø et al. [29]	Denmark	High	736	Part of the Copenhagen Prospective Studies on Asthma in Childhood (COSPAC2010) cohort	Nil	Infant gut microbiota
Hurtado et al. [44]	Spain	High	291	Healthy pregnant women who received preventative antibiotics 48 hours before/after childbirth and had the intention to breastfeed for 16 weeks	Nil	Breastmilk microbiota
Ismail et al. [30]	Australia	High	98	Doctor-diagnosed allergy	Asthma, eczema, food allergy and allergic rhinitis	Infant gut microbiota
Jiménez et al. [42]	Spain	Medium	20	Clinical symptoms of staphylococcal mastitis including; breast pain, redness, flu symptoms (fever > 38.5), milk staphylococcal count higher than 4 log_10_ CFU/mL and milk leukocyte count higher than 6 log_10_ CFU/mL. Women received antibiotics for 2–4 weeks, but the antibiotic (which was completed 2 weeks before the study) did not improve their symptoms	Staphylococcal mastitis	Breastmilk microbiota
Jinno et al. [31]	Japan	Medium	84	Healthy pregnant women. History of allergic disease was included	Not restricted to history of allergic disease *	Infant gut microbiota
Kamng’ona et al. [18]	Malawi	Medium	869	Ultrasound confirmed pregnancy < 20 weeks’ gestation	Nil	Infant gut microbiota
Korpela et al. [32]	Finland	High	1223	At least one parent had diagnosed allergic disease	Asthma, allergic rhinitis, atopic eczema	Infant gut microbiota
Kortekangas et al. [19]	Malawi	Medium	631	See Kamng’ona et al. [18]	Nil	
Lahtinen et al. [33]	Australia	High	122	Part of the Prevention of eczema in infants at high risk of developing allergic diseases study	Infants had increased risk of allergic disease	Infant gut microbiota
Maldonado-Lobón et al. [43]	Spain	Medium	148	Painful breastfeeding and milk bacterial counts >3 log_10_ CFU/mL	Painful breastfeeding not associated with acute mastitis	Breastmilk microbiota
Mastromarino et al. [45]	Italy	Medium	66	Healthy pregnant women	Nil	Breastmilk microbiota
Murphy et al. [34]	New Zealand	High	600	Mother or biological father had a history of allergic disease	Asthma, eczema, hay fever	Infant gut microbiota
Rinne et al. [36]	Finland	High	96	One close relative (mother, father or sibling) with allergic disease	Atopic dermatitis, allergic rhinitis, asthma	Infant gut microbiota
Rinne et al. [35]	Finland	High	132	See Rinne et al. [36]		Infant gut microbiota
Rutten et al. [37]	Netherlands	High	123	Pregnant women with diagnosis of allergic diseases or families in which the biological father, as well as at least 1 sibling, suffers from allergic disease	Atopic eczema, food allergy, asthma, allergic rhinitis	Infant gut microbiota
Shadid et al. [46]	Germany	Medium	48	Healthy pregnant women with an uncomplicated pregnancy and aiming for a vaginal delivery	Nil	Infant gut microbiota
Simpson et al. [38]	Norway	Medium	252	See Avershina et al. [25]		Breastmilk microbiota
Sordillo et al. [23]	USA	High	880	*Part of the V*itamin D Antenatal Asthma Reduction Trial (VDAART). Infants with a maternal history or biological father history of allergic disease	Asthma, eczema, allergic rhinitis	Infant gut microbiota
Wickens et al. [39]	New Zealand	High	423	Woman or biological father had a history of allergic disease. Healthy women intending to breastfeed	Asthma, eczema, hay fever	Breastmilk microbiota
Wickens et al. [40]	New Zealand	High	474	See Wickens et al. [39]		Infant gut microbiota

* Participants in the study may or may not have a FH of allergic disease. A FH of allergic disease was not part of the exclusion criteria. ^ The study did not state their exclusion criteria as part of their methodology.

**Table 2 nutrients-13-01137-t002:** Summary of results of the effects of maternal nutritional supplementation on the infant gut microbiota.

Reference	Intervention			Main Outcome	Outcome Observed (Yes/No)	Main Finding
	Supplement	Timing	Duration			
Abrahamsson et al. [24]	Probiotic (single strain)	Pregnancy	36 weeks’ gestation–12 months postpartum	Bacterial counts	Yes	Higher counts of *Bifidobacteria* in the infant stool in the intervention group; highest at 5–6 days after birth (82% in the treated vs. 20% in the placebo group, *P* < 0.001).
Avershina et al. [25]	Probiotic (multiple strain)	Pregnancy and lactation	36 weeks’ gestation–3 months postpartum	Bacterial counts	Yes	Children presenting with symptoms of atopic dermatitis not prevented by probiotic treatment have a different microbiota with an increased number of *Bifidobacterium dentium* (*P* = 0.001).
Baldassarre et al. [20]	Probiotic (multiple strain)	Pregnancy and lactation	36 weeks’ gestation–4 weeks postpartum	Bacterial counts	Yes	Number of *Lactobacilli* in the faeces of neonates from the probiotic group was higher than the control (*P* < 0.05).
Dotterud et al. [26]	Probiotic (multiple strain)	Pregnancy and lactation	36 weeks’ gestation–3 months postpartum	Bacterial diversity and colonization of the probiotic	Yes (colonization), no (diversity)	No change in the bacterial diversity. At 10 days and 3 months postpartum, both the prevalence and abundance of *Lactobacillus rhamnosus* (*P* < 0.005) were significantly increased in the infant stool.
Fonollá Joya et al. [22]	Probiotic (single strain)	Lactation	16 weeks of intervention whilst breastfeeding	Bacterial counts	Yes	Significant correlation was observed in the load of *Lactobacillus, Staphylococcus*, *Bacteroides* and *Escherichia coli* present in the infant faeces of supplemented mothers (*P* < 0.05).
Grönlund et al. [27]	Probiotic (multiple strain)	Pregnancy and lactation	2 months before delivery–2 months breastfeeding	Colonization of the probiotic	Yes (after cessation of supplement), no (during supplementation)	Association between maternal probiotic treatment and infant gut was non-significant during supplementation (*P* = 0.11 for *Bifdobacterium* and *P* = 0.40 for *Bifidobacterium longum*) but significant after cessation of supplementation (*P* = 0.043 for *Bifidobacterium* and *P* = 0.023 for *Bifidobacterium longum*)
Grześkowiak et al. [28]	Probiotic (multiple strain)	Pregnancy and lactation	2 months before delivery–2 months breastfeeding	Bacterial counts	Yes	Higher percentages of faecal *Lactobacillus/Enterococcus* (*P* < 0.003) and lower *Bifidobacteria* levels (*P* = 0.018) were detected in the intervention group compared to the control group.
Hjelmsø et al. [29]	Vitamin D3	Pregnancy	24 weeks gestation–1 week postpartum	Bacterial diversity	No	No significant differences were observed between the vitamin D supplementation group and the control group for the bacterial diversity at 1 week, 1 month or 1 year postpartum (*P* = 0.955, *P* = 0.865, *P* = 0.971).
Ismail et al. [30]	Probiotic (single strain)	Pregnancy	36 weeks’ gestation–delivery	Bacterial diversity	No	Supplementation did not alter the mean number of peaks in the infant faeces (AluI 14.4 vs. 15.5, *P* = 0.17, 95% CI -0.4, 2.5; Sau96I 17.3 vs. 15.8, *P* = 0.15, 95% CI -3.5, 0.5).
Jinno et al. [31]	Prebiotic	Pregnancy and lactation	26 weeks’ gestation–1 month postpartum	Bacterial counts	Yes (*Bifidobacterium longum*) and no (*Bifidobacterium)*	No significance in the number of *Bifidobacterium* in the intervention group compared to the control (*P* = 0.50), however, increased numbers of *Bifidobacterium longum* in the intervention group were seen (*P* = 0.01).
Kamng’ona et al. [18]	LNS	Pregnancy and lactation	Pregnancy–6 months postpartum	Bacterial diversity	Yes	Higher alpha diversity (Shannon index *P* = 0.032), Pielou’s evenness function (*P* = 0.043), and increased species richness (*P* = 0.08) at 18 months in infants of mothers in the intervention group.
Korpela et al. [32]	Probiotic (multiple strain)	Pregnancy	35 weeks’ gestation–delivery	Bacterial counts	Yes	Infants of mother’s supplemented and also breastfed had a twofold increase in abundance of *Lactobacilli*, and a 29% increase of *Bifidobacteria* (*P* < 0.0001).
Kortekangas et al. [19]	LNS or MMN	Pregnancy and lactation	Pregnancy–6 months postpartum	Bacterial diversity	No	A higher microbiota maturity and diversity at 6 months was associated with a lower incidence rate of fever in the following 6 months (*P* < 0.007 and *P* < 0.031, respectively).
Lahtinen et al. [33]	Probiotic (single strain)	Pregnancy	36 weeks’ gestation–delivery	Bacterial counts	Yes (*Bifidobacterium longum* and *breve)*, no (*Bifidobacterium adolescentis* and *angulatum*)	*Bifidobacterium longum* group was detected more frequently in the probiotic group (*P* < 0.01; prevalence ratio, 1.35; 95% CI, 1.06-1.72). An increased prevalence of *Bifidobacterium breve* was also seen (prevalence ratio, 1.39; 95% CI, 0.88-2.21) however, there was a decreased prevalence of *Bifidobacterium adolescentis* (prevalence ratio, 0.64; 95% CI, 0.35-1.19) and *Bifidobacterium angulatum* (prevalence ratio, 0.68; 95% CI, 0.30-1.53) in the probiotic group.
Murphy et al. [34]	Probiotic (single strain)	Pregnancy and lactation	35 weeks’ gestation–6 months postpartum	Bacterial diversity and colonization of the probiotic	Yes (colonisation), no (diversity)	No significant differences in bacterial diversity (Bray–Curtis distance) (*P* > 0.05). *Lactobacillus rhamnosus* DNA was detected almost exclusively in participants in the *Lactobacillus rhamnosus* probiotic group. *Bifidobacterium lactis* DNA was observed in all groups and was most abundant in the *B. lactis* group overall, (*P* = 2.1×10^13^, Kruskal–Wallis test).
Rinne et al. [36]	Probiotic (single strain)	Pregnancy and lactation	36 weeks’ gestation–6 months postpartum	Bacterial counts	No	Total numbers of bacteria in faecal samples decreased from 3 to 12 months of age; (*P* < 0.0001). Numbers were comparable between probiotic and placebo groups; (*P* = 0.70). *Bifidobacterium* counts followed a decreasing trend in both control and probiotic groups, (*P* < 0.0001 and *P* < 0.0001)
Rinne et al. [35]	Probiotic (single strain)	Pregnancy and lactation	36-38 weeks’ gestation–6 months postpartum	Bacterial counts	Yes (*Clostridia*), no (*Bifidobacteria, Bacteroides* and *Lactobacillus/Enterococcus)*	No differences in *Bifidobacteria, Bacteroides* or *Lactobacillus/Enterococcus* species at 6 months between the intervention and control groups (*P* = 0.145, *P* = 0.882, *P* = 0.817, respectively). At 6 months, there were less *Clostridia* in the faeces of the control compared with the probiotic group (*P* = 0.026). At 2 years, there were less *Lactobacilli/Enterococci* and *Clostridia* in the faeces of the probiotic group (*P* = 0.011 and *P* = 0.032, respectively).
Rutten et al. [37]	Probiotic (multiple strain)	Pregnancy and lactation	34 weeks’ gestation–1yr postpartum	Bacterial diversity and colonization of the probiotic	Yes	Diversity of *Bacteroidetes* was higher after two weeks in the placebo group. Detection of *Bifidobacteria* was higher at 1 month (*P* = 0.003) and *Lactococcus lactis* was higher at 2 weeks of age (*P* = 0.001), and 1 month (*P* = 0.03) in the probiotic group.
Shadid et al. [46]	Prebiotic	Pregnancy	25 weeks’ gestation–delivery	Bacterial counts	No	*Bifidobacteria* counts in the maternal gut were significantly higher in the supplemented group (*P =* 0.026), however, no significance was observed in the infant gut.
Sordillo et al. [23]	Vitamin D3	Pregnancy	10-18 weeks’ gestation–delivery	Bacterial diversity	No	No significant differences between vitamin D supplementation on either the alpha diversity or beta diversity of the infant gut microbiota (*P* > 0.05).
Wickens et al. [40]	Probiotic (single strain)	Pregnancy and lactation	35 weeks’ gestation–6 months postpartum	Colonization of the probiotic	Yes	Probiotic group had increased detection rates for the probiotic in faecal samples at 3, 12, and 24 months of age (*P* < 0.0001).

**Table 3 nutrients-13-01137-t003:** Summary of results of the effects of maternal nutritional supplementation on the breastmilk microbiota.

Reference	Intervention			Main outcome	Outcome Observed (Yes/No)	Main Finding
	Supplement	Timing	Duration			
Abrahamsson et al. [24]	Probiotic (single strain)	Pregnancy	36 weeks’ gestation–12 months postpartum	Colonization with the probiotic	Yes	Prevalence of *Lactobacillus reuteri* was higher in the colostrum compared to mature breastmilk in the intervention vs. control group (12% vs. 2%, *P* = 0.002).
Baldassarre et al. [20]	Probiotic (multiple strain)	Pregnancy and lactation	36 weeks’ gestation–4 weeks Postpartum	Bacterial counts	Yes	*Lactobacilli* and *Bifidobacteria* counts were higher in the colostrum of the intervention group (*P* = 0.099 and *P* < 0.05).
Dewanto et al. [21]	Probiotic (single strain)	Pregnancy and lactation	Enrolment in the study–4 months postpartum	Colonization with the probiotic	Yes	14% of supplemented women were positive for the probiotic at delivery, and 20% were positive at 3 months postpartum.
Fernández et al. [41]	Probiotic (single strain)	Pregnancy	30 weeks’ gestation–delivery	Bacterial counts and colonization with the probiotic	Yes	Small (0.19 [95% CI, 0.09–0.30] log_10_ CFU/mL) but significant difference was observed (*P* < 0.001) in the mean bacterial counts of women receiving the probiotic compared to those in the control. The probiotic was detected in the breastmilk of 59% of supplemented women and 12.5% of the placebo group.
Fonollá Joya et al. [22]	Probiotic (single strain)	Lactation	16 weeks of intervention whilst breastfeeding	Presence of bacterial species	Yes	A significant correlation was observed between supplementation and the breastmilk load of *Lactobacillus*, *Staphylococcus* and *Streptococcus* (*P* < 0.05).
Hurtado et al. [44]	Probiotic (single strain)	Lactation	16 weeks after delivery	Bacterial counts	Yes	In healthy women, lower levels of *Staphylococcus* subspecies were seen in the probiotic group (-48%; *P* = 0.013). The effect was also seen in cases of mastitis, with lower levels of *Staphylococcus* in the probiotic group (-58%. *P* = 0.065). *Staphylococcus* species load was lower in the breastmilk of women in the probiotic group (*P* = 0.025) at the end of the intervention.
Jiménez et al. [42]	Probiotic (multiple strain)	Lactation	30 days of treatment	Bacterial counts	Yes	On day 30 of treatment, the mean *Staphylococcal* count in the probiotic group (2.96 log10 CFU/mL) was lower than the control group (4.79 log10 CFU/mL), (*P* = 0.002).
Lahtinen et al. [33]	Probiotic (single strain)	Pregnancy	36 weeks’ gestation–delivery	Colonization with the probiotic	Yes	At birth, 66.7% of mothers in the probiotic group were colonized, compared to 11.8% in the placebo (*P* < 0.001; prevalence ratio, 5.67; 95% CI, 2.19–14.64).
Maldonado-Lobón et al. [43]	Probiotic (single strain)	Lactation	3 weeks	Bacterial counts	Yes	A significant decrease in *Staphylococcus* load in the probiotic group (*P* = 0.045) was seen on completion of the study. Probiotic supplementation showed a significant decrease (*P* = 0.011) in bacterial load when treated with 3 × 10^9^ CFU.
Mastromarino et al. [45]	Probiotic (multiple-strain)	Pregnancy and lactation	36 weeks’ gestation–4 weeks postpartum	Bacterial counts colonization with the probiotic	Yes	Counts of *Bifidobacteria* were significantly higher in the colostrum (median 1.7 × 104 cells/mL) and mature milk (median 1.4 × 104 cells/mL) of supplemented mothers. 3 mothers were colonised with the *Lactobacillus* species present in the probiotic in the placebo group, compared with 7 mothers in the treatment group.
Simpson et al. [38]	Probiotic (multiple strain)	Pregnancy and lactation	36 weeks’ gestation–3 months postpartum	Colonization with the probiotic and bacterial diversity	Yes (colonization), No (diversity)	8 women in the probiotic group and 1 woman in the placebo group had detectable levels of the administered bacteria. Probiotics had no statistically significant effect on the alpha or beta diversity of the breastmilk microbiota.
Wickens et al. [39]	Probiotic (single strain)	Pregnancy and lactation	12-16 weeks’ gestation–6 months postpartum	Colonization with the probiotic	No	The probiotic could not be detected in the breastmilk of supplemented women.

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
