# Peer review of "Impact of Maternal Nutritional Supplementation during Pregnancy and Lactation on the Infant Gut or Breastmilk Microbiota: A Systematic Review"

_nutrients, 2021, doi:10.3390/nu13041137_

Round 1

Reviewer 1 Report

Major Suggestions

  1. In the background section, the evidence on infant gut seeding beginning in utero is lacking information on meconium. Many studies have found that meconium does not have any bacteria, going against the in utero theory of infant microbiome inoculation. Please consider adding some insight into the meconium, and how this body of evidence feeds into your argument.
  2. Why was breastfeeding and human milk not considered as a search term? If you include these, does it change how many studies are identified?
  3. Since you included participants irregardless of health status, how would this bias your results? Please include this in the discussion of your paper, as I suspect this could greatly bias your results due to the large differences in current immune function, as one example.
  4. Something that I am really stuck on is why you didn’t include only healthy women. Please include a much more detailed response into why this wasn’t the case, as there is a lot of differing characteristics between healthy and unhealthy women. There is even a difference in breastmilk composition between these two groups of women. This is a major limitation that will need serious convincing among readers.
  5. The title is misleading. Since a lot of your included studies do not include breastmilk samples, it will be impossible to tell the actual impact on breastmilk composition. Please consider revising your title.

Minor Suggestions

  1. I would consider adding diet as a keyword to help increase your papers reach
  2. Please remove the first “the” in the following sentence, “Finally, the breastmilk and the process of breastfeeding”.
  3. Please make the following sentence more clear, and include how breastmilk and/or breastfeeding plan a role. “However, whether 78 alterations of the gut microbiota can be modulated through maternal diet, or maternal 79 nutritional supplementation remains unclear.
  4. Why was a third reviewer not included to resolve any disagreements?
  5. In the Data extraction and analysis section, why was date not included in the data extraction table? Please consider adding this information.
  6. In the statistical analysis section, were confidence intervals calculated? Please include this information here.
  7. What other sources were used in the identification of articles. Since only 2 were found, it would not be too much extra information to include what these sources were.
  8. Figure 2 appears as random numbers and I am not sure what this is representing. Perhaps the figure was not uploaded correctly. Please double check.
  9. Please consider making Figure 3 into two separate figures. Right now there is figure 3 a and b, and then another section of figure c further down. This is a little confusing. Also, for the heat map, I would really suggest using a different color scheme. It is not intuitive the way it is right now, and to be frank, I am unsure what exactly you are trying to show.

Reviewer 2 Report

Manuscript details:
Journal: Nutrients
Manuscript ID: nutrients-1143769

Type of manuscript: Review
Authors: Aneesa Z. Zaidi, Sophie E. Moore, and Sandra G. Okala

Dear authors,

Your systemic review entitled “Impact of maternal nutritional supplementation during pregnancy and lactation on the breastmilk and infant gut microbiota: a systematic review” is comprehensive and nicely written. In this article, the authors investigated the effect of maternal nutritional supplementation on the breastmilk or infant gut microbiota and suggested that maternal supplementation using probiotics during pregnancy and lactation period affects bacterial colonization in breastmilk and infant's gut. Furthermore, it may results in altered bacterial count in breast milk and the infant's gut. The article is well written, looks scientifically sound and backed-up by an impressive number of references.

However, I would like the authors to address a few minor corrections in the systematic review.

  • On page 5, in Figure 2, I don’t see any images but only a few numbers and names (for example Europe Americas Western Pacific South-East Asia Africa), please correct.
  • On page 3, in lines 144-145, it says “Each type of bias was graded as low, high or unclear based on provided evidence. The GRADE approach [14] was also used to determine the overall quality of each study”. Please mention more details about the GRADE approach and how the quality of each study was determined?
  • In the article, the author has mentioned EBF and FH (in Table 1 and in the text). I assume it is for exclusive breastfeeding, please specify the complete form of EBF and FH at the time of first use followed by abbreviation?
  • On page 11, in lines 249-251, at the end of the intervention has been written twice, please correct.
  • In line 227 women has been written twice please correct.
  • On page 12, in lines 291-299, the author says the significant increase in Bifidobacteria counts at one month of the infant’s age (P = 0.0033), and then it says there was no evidence of infant gut colonization during supplementation (P = 0.11 for Bifidobacterium and P = 0.40 for Bifidobacterium longum), however, after cessation of the supplement, evidence of probiotic colonization in the infant gut couple be detected (P = 0.043 for Bifidobacterium and P = 0.023 for Bifidobacterium longum). Please rephrase it clearly.
  • On page 17, in line 365 during pregnancy has been written twice. Please correct.
  • On page 17, in lines 362-369, it seems that the infants in the LNS group where LNS was supplemented to women during pregnancy until 6 months postpartum, were also given LNS from 6 to 18 months of age, but it is not clear if in other groups where women were supplemented with either multiple micronutrients (MMN), iron and folic acid (IFA) tablet-based supplements if the infants were also supplemented with the same or not?. Please rephrase clearly.

Thank you

Round 2

Reviewer 1 Report

Thank you for the thorough responses to each comment. I think the manuscript is much approved.